# (Photo)toxicity of Partially Oxidized Docosahexaenoate and Its Effect on the Formation of Lipofuscin in Cultured Human Retinal Pigment Epithelial Cells

**DOI:** 10.3390/antiox13111428

**Published:** 2024-11-20

**Authors:** Linda M. Bakker, Michael E. Boulton, Małgorzata B. Różanowska

**Affiliations:** 1School of Optometry and Vision Sciences, Cardiff University, Cardiff CF24 4HQ, Wales, UK; lmbakker1@hotmail.com; 2Department of Ophthalmology and Visual Sciences, University of Alabama at Birmingham, Birmingham, AL 35294, USA; meboulton@uabmc.edu; 3Cardiff Institute for Tissue Engineering and Repair (CITER), Cardiff University, Cardiff CF10 3AX, Wales, UK

**Keywords:** docosahexaenoic acid, omega-3 fatty acid, retina, lipid peroxidation, retinal pigment epithelium, photoreceptor outer segments, (photo)toxicity, apoptosis, lipofuscin, zeaxanthin, α-tocopherol, age-related macular degeneration, AREDS2

## Abstract

Docosahexaenoate is a cytoprotective ω-3 polyunsaturated lipid that is abundant in the retina and is essential for its function. Due to its six unsaturated double bonds, docosahexaenoate is highly susceptible to oxidation and the formation of products with photosensitizing properties. This study aimed to test on cultured human retinal pigment epithelial cells ARPE-19 the (photo)cytotoxic potential of partly oxidized docosahexaenoate and its effect on the formation of lipofuscin from phagocytosed photoreceptor outer segments (POSs). The results demonstrate that the cytoprotective effects of docosahexaenoate do not counteract the deleterious effects of its oxidation products, leading to the concentration-dependent loss of cell metabolic activity, which is exacerbated by concomitant exposure to visible light. Partly oxidized docosahexaenoate does not cause permeability of the cell plasma membrane but does cause apoptosis. While vitamin E can provide partial protection from the (photo)toxicity of partly oxidized docosahexaenoate, zeaxanthin undergoes rapid photodegradation and can exacerbate the (photo)toxicity. Feeding cells with POSs enriched in partly oxidized docosahexaenoate results in a greater accumulation of intracellular fluorescent lipofuscin than in cells fed POSs without the addition. In conclusion, partly oxidized docosahexaenoate increases the accumulation of lipofuscin-like intracellular deposits, is cytotoxic, and its toxicity increases during exposure to light. These effects may contribute to the increased progression of geographic atrophy observed after long-term supplementation with docosahexaenoate in age-related macular degeneration patients.

## 1. Introduction

Docosahexaenoic acid is a ω-3 polyunsaturated fatty acid with six unsaturated double bonds, which is highly abundant in the lipid membranes of the retina, particularly in photoreceptor outer segments (POS), where it accounts for 32% of free fatty acids and fatty acyl chains of phospholipids [1]. Docosahexaenoate is essential for the function of the retina, enabling visual transduction cascade and, thereby, sight [2,3,4,5,6,7,8,9]. Moreover, the products of enzymatic oxidation of docosahexaenoate have been shown to play a protective role for photoreceptors and retinal pigment epithelium (RPE) [10,11,12,13,14,15,16,17]. Several lipid mediators that can be enzymatically synthesized from docosahexaenoic acid have been postulated to be specialized for resolving inflammation, although it is still unclear whether they play a role in human physiology [18]. This, together with epidemiological evidence suggesting that higher dietary intake of docosahexaenoate is associated with a lower risk of development of age-related macular degeneration (AMD), which is the primary cause of vision loss in people above 60 years of age in developed countries, led to clinical testing of supplementation with docosahexaenoate as a potential treatment for slowing down the progression of AMD, as well as in other retinal degenerative diseases, including Stargardt’s disease and retinitis pigmentosa [19,20,21,22,23,24,25,26,27,28,29,30,31,32]. However, most of these clinical trials have not demonstrated a beneficial effect of supplementation with docosahexaenoate on the development or progression of retinal diseases.

A potential explanation for the lack of the beneficial effects of docosahexaenoate supplementation on the progression of these retinal degenerations, which are associated with increased oxidative stress, is its oxidation. In the retina, docosahexaenoate is at risk of oxidation due to the high oxygen tension in POS, exposure to light in the presence of potent photosensitizers, namely retinaldehyde and lipofuscin, which, upon photoexcitation with ultraviolet or blue light, can photosensitize the generation of reactive oxygen species, such as singlet oxygen and superoxide radicals [33]. Singlet oxygen can oxidize unsaturated fatty acids, forming lipid hydroperoxides [34]. Superoxide can dismutate into hydrogen peroxide. The hydrogen peroxide becomes particularly damaging upon interaction with reduced metal ions, such as iron. In the AMD retina, there is an increased level of iron in the retinal photoreceptors and RPE, including easily chelateable iron, in comparison with age-matched healthy retina [35,36]. Reduced iron ions can decompose hydrogen peroxide, resulting in the formation of the highly reactive hydroxyl radical, which can initiate a chain of lipid peroxidation by abstracting hydrogen from unsaturated fatty acids [34]. Reduced iron ions can also decompose lipid hydroperoxides to form lipid-derived radicals, each of which can also initiate a chain of lipid peroxidation. What follows is the propagation of the lipid peroxidation cascade, resulting in the formation of numerous lipid peroxides. These processes are all conducive to docosahexaenoate oxidation. Lipid peroxides can decompose into reactive derivatives, including aldehydes and Michael acceptors, which can react with proteins and nucleic acids [37]. Indeed, exposure of rodents to intense light results in the formation of adducts between proteins and aldehydes derived from the oxidation of ω-6 and ω-3 polyunsaturated fatty acids, 4-hydroxynonenal and 4-hydroxyhexenal, respectively, and the photoreceptor loss, and both – the adducts accumulation and photoreceptor loss are increased in retinas with increased amounts of docosahexaenoate [38]. Moreover, adducts of docosahexaenoate oxidation products have been found in photoreceptors and RPE, especially in RPE lipofuscin, and also in drusen and other deposits on the basolateral surface of the RPE cell layer, as well as in Bruch’s membrane, which separates RPE from the choroidal blood supply [39,40,41]. Therefore, it has been suggested that oxidized lipids can contribute to the pathogenesis of AMD [42,43], but it is still unclear what mechanisms can be involved.

Several studies in vitro have demonstrated that exposure of cultured RPE cells to docosahexaenoate under conditions that favor lipid oxidation or to a specific product of docosahexaenoate oxidation leads to damaging effects, including cell death [44,45,46,47,48,49,50,51]. Moreover, it has also been demonstrated that docosahexaenoate, oxidized by exposure to air, forms chromophores absorbing UVB, UVA, and visible light, which exhibit photosensitizing properties resulting in the production of singlet oxygen and superoxide when oxidized docosahexaenoate is exposed to light [52]. It is not clear whether the (photo)cytotoxic properties of oxidized docosahexaenoate can play a deleterious role in the presence of intact docosahexaenoate or whether the cytoprotective effects of unoxidized form can counteract its deleterious effects. Therefore, the first aim of this study was to evaluate the effects of partly oxidized docosahexaenoate on the metabolic activity and viability of cultured human RPE cells ARPE-19 in the dark and with concomitant exposure to visible light.

Among the RPE roles that are essential for the survival and proper functions of photoreceptors is the daily phagocytosis of distal tips of POSs [33]. Incomplete lysosomal degradation of phagocytosed POSs tips leads to the accumulation of residual bodies in the cytoplasm called lipofuscin. Several studies have demonstrated that increased oxidation of POSs or inclusion in POSs of certain products of lipid oxidation, such as 4-hydroxynonenal, a product of oxidation of ω-6 polyunsaturated fatty acids, leads to an increase in lipofuscin formation. However, it is not clear if oxidized docosahexaenoate can also increase lipofuscin formation from phagocytosed POSs. Therefore, the second aim of this study was to determine the effects of phagocytosis of POSs enriched in partly oxidized docosahexaenoate on the accumulation of lipofuscin-like material in cultured RPE cells.

The results demonstrate that the neuroprotective effects of docosahexaenoate do not counteract the deleterious effects of its oxidized form, leading to the concentration-dependent loss of cell metabolic activity, which is exacerbated by concomitant exposure to visible light. Partly oxidized docosahexaenoate does not cause significant acute damage to the cell plasma membrane but does cause apoptotic cell death. While vitamin E can provide partial protection from the toxic effects of partly oxidized docosahexaenoate, zeaxanthin undergoes rapid photodegradation and can exacerbate (photo)toxicity. Enrichment of POSs with partly oxidized docosahexaenoate results in a greater accumulation of intracellular lipofuscin-like deposits than in cells fed POSs without the addition of partly oxidized docosahexaenoate. Extended feeding of cells with POSs enriched with partly oxidized docosahexaenoate results in cell death. In conclusion, partly oxidized docosahexaenoate is cytotoxic to ARPE-19 cells, and its toxicity increases during exposure to light and can increase the accumulation of lipofuscin.

## 2. Materials and Methods

### 2.1. Materials

Phosphatidylcholine with two docosahexaenoyl chains, 1,2-didocosahexaenoyl-sn-glycero-3-phosphocholine (22:6PC), was purchased from Avanti Polar Lipids, Alabaster, AL, USA; cat #850400C (supplied as a solution in chloroform). Zeaxanthin was obtained from DSM Nutritional Products AG (Basel, Switzerland). Streptomycin, kanamycin, penicillin, vitamin E (α-tocopherol), trypsin, ethylenediaminetetraacetic acid (EDTA), 3-(4,5-dimethylthiazol-2-yl)-2,5-diphenyl tetrazolium bromide (MTT), propidium iodide, dimethyl sulfoxide (DMSO), hydrochloric acid, acetic acid, glutaraldehyde, osmium tetroxide, and imidazole were purchased from Sigma-Aldrich (Gillingham, UK). Chloroform, methanol, acetonitrile, isopropanol, ethanol, and Hoechst 33342 were purchased from Fisher Scientific, Loughborough, UK. Gas cylinders with carbon dioxide and Pureshield argon were from BOC Ltd. (Woking, UK). A 1:1 mixture of Dulbecco’s modified Eagle’s medium and Ham’s F12 medium (DMEM:F12), Dulbecco’s PBS (D-PBS), and Fungizone (Amphotericin B) were purchased from GIBCO Ltd, London, UK. Fetal calf serum (FCS) was purchased from Bio-West, UK. Betadine solution was from AAH Pharmaceuticals Ltd., Walsgrave Triangle, Coventry, UK. Agar was from Difco Laboratories, Detroit, MI, USA.

A spontaneously immortalized human RPE cell line, ARPE-19, was purchased from the American Type Culture Collection (ATCC) through LGC Promochem, UK. The ApopTag^®^ Plus Rhodamine In Situ Apoptosis Detection Kit apoptosis assay was purchased from Chemicon International, Temecula, CA, USA. Bovine eyes were purchased from a local abattoir.

### 2.2. Preparation of 22:6PC Lipid Vesicles (Liposomes)

To prepare liposomes, firstly, a lipid film of 22:6PC was formed using a rotary vacuum evaporator (Buchi, Flawil, Switzerland) [52]. All procedures were conducted under a dim red light. A freshly opened solution of 22:6PC in chloroform was transferred to a round bottom flask and attached to the rotary evaporator, and the system was flushed with argon. The flask was maintained at 37 °C, partly immersed in a water bath, and rotated while the pressure was gradually decreased to 40 mbar as the solvent evaporated. To ensure the lipid film was dry and solvent-free, the evaporated chloroform was removed from the waste container, the system was flushed with argon, and the pressure was again reduced to 40 mbar and left for 90 min in the dark at room temperature. To form liposomes, phosphate-buffered saline (PBS) was added gradually to the flask containing the lipid film, and the film was rehydrated using a combination of rotation, vortexing, and warming to 37 °C. The final 22:6PC concentration was 50 mM. Samples that were to be kept unoxidized were saturated with argon for at least 30 min to remove remnants of air/oxygen, then stored tightly capped at −80 °C.

To ensure the formation of uniformly sized liposomes, liposomes were aliquoted into cryovials and put through 5 freeze/thaw cycles (freezing in liquid nitrogen, then thawing in a 37 °C water bath). Before use in experiments with cultured cells, liposomes were extruded using a Liposofast 100 J extruder (Glen Creston Ltd., London, UK) with two 200 nm filters powered by compressed argon to ensure sterility and uniform size. Liposomes used for experiments on cultured ARPE-19 cells contained partly oxidized 22:6PC with 70–75% intact docosahexaenoate remaining.

### 2.3. Oxidation of 22:6PC

To oxidize 22:6PC, the flasks with the lipid film or liposomes were incubated on a heating block at 37 °C in the dark, with containers opened weekly to allow replenishment of oxygen [52]. To ensure that oxidation occurred as evenly as possible within each liposome batch, equal aliquots were put into identical tubes. The lipid film was kept in a dry environment on the inner surface of a spherical rotary evaporator flask.

The first batch of liposomes and the lipid film were allowed to oxidize simultaneously for a total of 175 days. At selected time points, aliquots of liposomal suspension were withdrawn to monitor the time course of lipid oxidation by absorption spectroscopy and high-performance liquid chromatography (HPLC), as described below. After 175 days of oxidation, the lipid film was hydrated to produce liposomes, and its oxidation state was measured by the same methods as for 22:6PC oxidized in liposomes.

### 2.4. Monitoring of 22:6PC Oxidation

Progress of oxidation was monitored by extracting lipids from the liposome suspension, followed by absorption spectroscopy using the U-2800 UV–VIS Hitachi Spectrophotometer with Hitachi UV Solutions 2.0 software (Tokyo, Japan) and high-performance liquid chromatography (HPLC; Waters 2695 Separations Module with Millennium 32 v. 4.0 software, Waters Ltd., Wilmslow, UK) as described below.

#### 2.4.1. Extraction of Lipids

Lipids were extracted from liposomes using a modified Folch’s procedure [53,54]: 0.5 mL liposome suspension in PBS (corresponding to 50 mM of 22:6PC used as a starting material) was added to a 0.8 mL chloroform/methanol mixture (2:1 *v*/*v*; both Fisher Scientific, Loughborough, UK) and vortexed briefly, then centrifuged for 1 min. This resulted in the formation of an upper polar phase enriched with methanol/PBS and a lower chloroform-enriched phase. The two phases were removed and analyzed separately. When removing extracted samples as described above, a small amount of each phase was left in the tube to prevent contamination from the other phase during removal. To maximize the total amount of sample extracted, the extractions were repeated twice more: the volume removed from each phase was replaced with methanol/PBS or chloroform-enriched phases obtained by mock extraction of PBS using the same method described for liposomes above (i.e., using PBS instead of liposome suspension). Samples were vortexed and centrifuged again; further samples were removed, and then the process was repeated.

#### 2.4.2. UV–Visible Absorption Spectroscopy

For liposomal components solubilized in the chloroform-enriched phase, the solvent was evaporated by passing argon gas over the sample while heating to 37 °C on a heating block in the dark. The dried lipid film was then dissolved in a mixture of 2 mL acetonitrile and 100 µL methanol, and spectra were measured.

For absorption spectra measurements after repeat extraction of the chloroform phase, the samples were dried under argon as above and re-dissolved in an acetonitrile and methanol mixture (20:1, *v*/*v*). Appropriate dilutions of the samples were carried out, and the spectra were re-measured to ensure the upper limit of absorbance was below 1.5 at the wavelengths of interest. This allowed the same sample to be used to monitor absorption at >400 nm (in the most concentrated samples) and to monitor the upper limit of the remaining unoxidized 22:6PC and the formation of markers of oxidation in more dilute samples.

#### 2.4.3. HPLC

For HPLC analysis, docosahexaenoate was hydrolyzed from the phosphatidylcholine headgroup [55,56]. After repeating the chloroform/methanol extractions and drying under argon, samples were saponified by re-suspending in 2 mL of argon-saturated potassium hydroxide in ethanol and incubating for 30 min at 100 °C. After cooling to room temperature, 1 mL of argon-saturated dH_2_0 and 100 µL of concentrated hydrochloric acid were added. The free docosahexaenoic acid was extracted twice with 3 mL of argon-saturated hexane and dried under argon.

Samples were then solubilized in 3:1 methanol/acetonitrile (*v*/*v*), filtered using Vectaspin microfilters (0.45 µm, polypropylene; Whatman, UK), and then analyzed by reverse phase HPLC using a Waters 2695 Separations Module with Millennium software (Waters, UK). Separation was carried out isocratically using an Ace 5 C8 column (150 mm × 4.6 mm, particle size of 5 µm; Hichrom Limited, Reading, UK), with a mobile phase consisting of acetonitrile/HPLC-grade water/HCl (90/10/0.05%, *v*/*v*/*v*) at a flow rate of 1 mL/min. The elution profile was monitored using a Waters 996 Photodiode Array Detector (Waters, UK). Before running samples, the column was equilibrated with mobile phase solvent until absorption of the eluent at 210 nm stabilized, then 10 µL of the sample was injected, and elution was monitored for 10 min at 190–600 nm.

### 2.5. Preparation of Lipid Samples Containing Lipophilic Antioxidants

To produce liposomes containing partly oxidized 22:6PC and lipophilic antioxidants, liposomes oxidized for 18 days were extracted by the modified Folch’s method described above. Zeaxanthin was solubilized in ethanol, and absorbance at 454 nm was measured to calculate the concentration using the molar absorption coefficient of 1.52 × 10^5^ M^−1^cm^−1^ [57]. An appropriate amount of α-tocopherol was weighed and solubilized in chloroform. Antioxidants in their respective solvents were then added at selected concentrations to the extracted partly oxidized 22:6PC, and then the lipid–antioxidant mixtures were dried to form a lipid film using a rotary evaporator as described above. Liposomes were then produced by hydration with D-PBS, saturation with argon for 20 min, and put through five freeze/thaw cycles before use. Samples were stored at −80 °C under argon.

Alternatively, zeaxanthin was added to liposomes immediately before the cell exposure by making its stock solution in DMSO and adding it directly to partly oxidized 22:6PC liposome suspensions. In these experiments, all wells contained 0.2% (*v*/*v*) DMSO either as a solvent for zeaxanthin or as a vehicle control.

### 2.6. Cell Culture

The effects of partly oxidized 22:6PC were tested on a spontaneously immortalized human RPE cell line ARPE-19 (American Type Culture Collection (ATCC)) [58,59,60]. Cells from passage numbers 24–28 were grown in a DMEM:F12 supplemented with antibiotics (100 μg/mL streptomycin, 100 μg/mL kanamycin, and 60 μg/mL penicillin), Fungizone (1.25 μg/mL), and 10% *v*/*v* heat-inactivated FCS until the cells reached confluence. FCS concentration was then reduced to 2% until cells were required for splitting or experiments. Cells were maintained at 37 °C in a humidified incubator containing 5% CO_2_ and 95% air, and the culture medium was changed every 3 days. Cell populations were initially built up in 75 cm^2^ flasks, then split into 25 cm^2^ flasks, 12- or 24-well plates, or 4-well chamber slides as required for experiments. To subculture cells, the culture medium was aspirated, and cells were rinsed with sterile PBS. Cells were detached by adding 0.25% (*w*/*v*) trypsin + 0.02% EDTA in PBS and incubating at 37 °C for 2 min. Trypsin was then deactivated by adding a culture medium with 10% FCS, and cells were transferred to new flasks or multi-well plates (at an approximate surface area ratio of 1:3 according to the supplier’s recommendations).

### 2.7. Exposure of Cells to Partly Oxidized 22:6PC and/or Light

To mimic the physiological conditions where RPE cells may be exposed to oxidized 22:6PC present in photoreceptor outer segments, cultured ARPE-19 cells were exposed to partly oxidized 22:6PC (with 70–75% of intact docosahexaenoate) in the form of externally applied lipid vesicles suspended in DPBS. Cells were exposed to liposomes prepared by oxidation of 22:6PC as lipid film for 175 days or as liposomes oxidized for 18 days, with 70 and 75% of intact docosahexaenoate remaining, respectively. The latter were used for experiments with antioxidants and experiments with phagocytosis of photoreceptor outer segments (POSs).

Exposure to partly oxidized 22:6PC with and without concomitant exposure to light was carried out outside the incubator for 1 h, after which the liposomal suspensions were removed, cells were washed twice with DPBS, and a culture medium containing 2% FCS was added. Then, the cells were either used immediately for a metabolic activity assay or returned to the incubator for a further incubation period, with assays performed later.

Light exposure was carried out with a Sol lamp (Hönle UV Ltd., Birmingham, UK) as described [61]. Heat and UV filters (Lee Heat Shield and #226 Lee U.V., respectively; Lee Filters, Andover, UK) were placed between the lamp and glass plate supporting the culture plate. Spectral irradiance at the area occupied by wells in cell culture plates was measured between 380 and 780 nm in each experiment using a handheld spectroradiometer (Specbos 1201 with JETI LiMeS software; Glen Spectra, Stanmore, UK). The light intensity was adjusted by changing the distance between the glass plate relative to the lamp and/or by adding individual UV and/or gray filters to act as additional neutral density filters so to achieve an irradiance of 15 mW/cm^2^ (with 2.94 mW/cm^2^ from the spectral range below 500 nm where the photosensitized effects of oxidized docosahexaenoate occur) [52]. The illuminance was 39.8 klx. The temperature was between 29 and 32 °C. For experiments on dark-maintained cells, culture plates were wrapped in black foil and placed on the glass plate to ensure the same temperature as that experienced by light-exposed cells. Aluminium foil was placed between the black foil and the glass plate to prevent excess heat absorption.

### 2.8. Cytotoxicity Assays

Cell viability was assessed by adherent cell density, assays of plasma membrane integrity, and metabolic activity.

Cell morphology of ARPE-19 cell monolayer was visualized using an Olympus IX70 inverted microscope configured for phase contrast imaging, and images were obtained using a Spot RT color CCD camera and Spot Advanced Software (Diagnostic Instruments Inc., Sterling Heights, MI, USA).

The integrity of the plasma membrane was tested either immediately following exposure to partly oxidized 22:6PC and/or light or after 2 h of recovery in a fresh culture medium. In short, cells were incubated for 15 min at 37 °C in 1 μg/mL propidium iodide in DPBS. Cells were co-labeled with 3 μg/mL Hoechst 33342 to allow total cell numbers to be counted. Cells were washed and imaged immediately using the Olympus IX70 inverted microscope described above, using filter sets optimized for blue emission for Hoechst (excitation filter transmitting light 330–385 nm; long-pass emission filter > 420 nm) and red emission for propidium iodide (excitation: 510–550 nm; long-pass emission filter > 590 nm).

The metabolic activity of cells was measured by MTT assay [59,60,62] and carried out either immediately or 24 h after exposure to partly oxidized 22:6PC and/or light. In short, cells were incubated with 0.5 mg/mL MTT for 1 h. The MTT solution was removed, cells were washed with PBS, and formazan crystals were dissolved in isopropanol acidified with 40 mM HCl. Absorbance at 590 nm was measured in a microplate reader (Multiskan Ascent, Labsystems, Helsinki, Finland). All MTT results from a culture plate were normalized to control cells in the same plate, which was subjected to identical light/dark exposure conditions and post-incubation period but without partly oxidized 22:6PC and expressed as a percentage. Independently, the metabolic activity of cells exposed to light was calculated as a percentage of the metabolic activity of control cells incubated in paired plates in the dark.

### 2.9. Apoptosis Assays

For apoptosis assays, the duration of post-exposure incubation was varied to ensure that crucial steps in the cascade of cell death events were not missed. The cells were then imaged using phase contrast and fluorescence microscopy to monitor changes in gross morphology and nuclear condensation (using Hoechst labeling). DNA cleavage was assessed by Terminal dUTP Nick-End Labeling (TUNEL) assay using the ApopTag^®^ Plus Rhodamine In Situ Apoptosis Detection Kit according to the instructions supplied with the kit [63]. In short, cells were grown on 4-well chamber slides and used for experiments at least a week after switching to medium with 2% FCS. After exposure to partly oxidized 22:6PC with or without the concomitant exposure to 15 mW/cm^2^ light, cells were returned to the incubator for 12 h, after which the culture medium was removed, and the cells were rinsed in PBS and then fixed in 1% paraformaldehyde for 10 min at room temperature, then washed with PBS twice. Post-fixation was carried out with 2:1 ethanol:acetic acid for 5 min at −20 °C. After washing with PBS, the equilibration buffer (supplied with the kit) was applied for 10 s, then removed, and the cells were incubated with working-strength TdT enzyme for 1 h at 37 °C. The working strength stop/wash buffer was added and left to incubate for 10 min at room temperature after 15 s of agitation. Then, the cells were washed with PBS, and working-strength anti-digoxigenin–rhodamine conjugate (affinity purified sheep polyclonal antibody) was added and left for 30 min at room temperature in the dark. Then, the cells were again washed with PBS. Vectashield mounting medium containing 3 μg/mL Hoechst 33342 (to label all cell nuclei) was added, and a coverslip was applied. Cells were imaged using a Leica DMRA2 upright microscope, and images were obtained using a Leica DC500 CCD camera and Leica QFluoro software (Leica, Milton Keynes, UK).

### 2.10. Isolation of Bovine Photoreceptor Outer Segments (POSs)

Retinas were isolated from fresh bovine eyes. Muscle and fat were removed from the eyeballs, and the eyes were washed in Betadine solution and transferred to a tissue culture hood for the remainder of the process. An incision was made in the orbit posterior to the cornea using a scalpel, and scissors were used to remove the anterior segment and vitreous. The retina was gently detached from the eye cup and cut at the optic nerve. Retinas were either stored at 4 °C and used within 24 h or frozen at −80 °C and thawed as required. Sterile conditions were maintained throughout. POSs isolation was carried out following a modified version of the procedure of Papermaster [64]. All solutions were filtered and pre-chilled, and ice was used to ensure samples were maintained at a temperature of ~4 °C. The homogenization medium was added to the retinas (15 mL per 25 retinas), and a Teflon hand-held homogenizer was passed through the mixture three times. The homogenate was transferred to a cold centrifuge tube, then back into the homogenizer tube, and homogenization was repeated. The homogenate was centrifuged at 1500× *g* for 4 min at 4 °C (U-32R Hettich Zentrifugen with 1617 rotor; Boeco, Hamburg, Germany).

The supernatant was decanted into centrifuge tubes, and the pellet was homogenized again as above with a further 15 mL homogenization buffer and recentrifuged. The total volume of supernatant after the two centrifugations was determined before transferring it to a cold 500 mL conical flask. Two equivalent volumes of 0.01 M tris-acetate buffer (ice cold, pH 7.4) were added gradually while stirring. The POSs were then pelleted by centrifugation at 1500× *g* for 4 min at 4 °C. The pellet was re-suspended in 1.10 g/mL sucrose solution and repeatedly passed through an 18- then 21-gauge needle, rapidly ejecting the solution against the wall of the plastic tube each time.

Sucrose gradients were prepared using 1.15 and 1.11 g/mL sucrose solutions carefully layered into ultracentrifuge tubes to ensure negligible mixing at the interface. The crude POSs suspension in 1.10 g/mL sucrose was then carefully layered onto the gradient. After centrifugation at 50,000× *g* for 30 min at 4 °C in an ultracentrifuge (Sorvall Ultra-Pro 80, Sorvall Products, L.P., Newtown, CT, USA), POSs were carefully removed from the 1.11/1.15 g/mL interface using a sterile Pasteur pipette. POSs were recovered by dilution in 0.01 M tris-acetate and centrifugation in the ultracentrifuge at 45,000× *g* for 20 min. The resultant POSs pellet was then re-suspended in D-PBS, and the concentration of POSs was determined in 100 times diluted stock suspension using a Bürker hemocytometer and optical microscopy with a 20× objective. After each isolation procedure, a small amount of POSs suspension was added to a cell culture flask with medium and monitored for at least 24 h to ensure no microbial contamination.

### 2.11. Supplementation of Cells with POSs with and Without Partly Oxidized 22:6PC

Before use for supplementation of cultured cells, POSs were diluted to the required concentration in the cell culture medium in the absence or presence of selected concentrations of partly oxidized 22:6PC in liposomes. To facilitate the insertion of partly oxidized 22:6PC into POSs, the suspensions of POsS and partly oxidized 22:6PC liposomes were sonicated for 15 min.

The cultured cells were supplemented either with 1 × l0^7^ POSs/mL [65] or POSs with partly oxidized 22:6PC at selected concentrations. The control cells were fed with the culture medium only. The suspensions of partly oxidized 22:6PC liposomes/POSs were applied to cells in a 24-well cell culture plate, and the medium was changed every other day, with partly oxidized 22:6PC liposomes/POSs included in each feed.

### 2.12. Flow Cytometry Analysis of Cell Fluorescence

After feeding ARPE-19 cells for up to 4 weeks with partly oxidized 22:6PC liposomes/POSs, cells were washed with PBS, then gently detached by trypsinization and transported immediately on ice in a dark container to the flow cytometer (BD FACSCalibur with CellQuest software, BD Life Sciences, UK). Cells taken up by the flow cytometer passed through a laser beam of 488 nm wavelength and forward and side scatter (to determine the size and granularity of the cells, respectively), and fluorescence emission data were collected.

Detectors and filters allowed the separation of emitted fluorescent light into three channels with transmission maxima corresponding to different wavelengths: 530 nm emission (FL1), 585 nm (FL2), and 650 nm (FL3). The sensitivity of the detectors was adjusted to ensure that the fluorescence signal of control cells was limited to the lowest decade of the fluorescence intensity histogram and that the most fluorescent cells did not exceed the limits of the 4-decade scale.

Before data collection, a gate was set on the forward scatter (FSC) vs. side scatter (SSC) plot to exclude cell debris. For each sample, data were collected until 10,000 events, assumed to be cells, were counted within this gate. For each sample, FSC, SSC, and FL1 were collected. Data analysis and presentation were carried out using WinMDI 2.8 software (The Scripps Research Institute, La Jolla, CA, USA).

### 2.13. Transmission Electron Microscopy (TEM)

After two weeks of supplementation with POSs or POSs enriched with partly oxidized 22:6PC, cells were detached by trypsinization and then centrifuged to form pellets. The pellets were fixed for 1 h at 4 °C in a mixture of glutaraldehyde and osmium tetroxide buffered with 0.05 M imidazole, pH 7.4 (1 part 2.5% glutaraldehyde + 2 parts of 1% osmium tetroxide, both in 0.05 M imidazole). The pellet was scraped from the side periodically to enable full penetration of the fixative. After fixation, the pellet was broken up, and samples were centrifuged. Samples were then embedded in 3% (*w*/*v*) agar (Difco Laboratories Inc, Detroit, MI, USA), which was first dissolved by heating and then allowed to cool before embedding the samples. Pellets were scraped from the side to ensure total immersion in the agar. After solidification of the agar, the samples were removed from the tubes and cut to remove excess agar from around the specimens and to provide several small pieces of agar-embedded pellet for each treatment. The pieces were then washed 4 times for 10 min each in distilled water to remove excess fixatives, then stained with 0.5% aqueous uranyl acetate for 60 min in the dark at 4 °C.

The specimens were dehydrated in a graded series of ethanol concentrations of 30%, 50%, 70%, 80%, 90%, and 100% for 10 min each, followed by two additional incubations in 100% ethanol. Then, the specimens were incubated in propylene oxide for 2 × 10 min. Samples were infiltrated overnight at room temperature with the resin composed of a 1:1 mixture of propylene oxide and epoxy resin Araldite CY212 kit (composed of 5 g araldite CY212, 5 g dodecenylsuccinic anhydride (DDSA), and 0.15 g benzyldimethylamine (BDMA); Agar Scientific Ltd., Stansted, UK). The following day, samples were transferred into fresh resin (i.e., Araldite mixture without propylene oxide) in plastic molds and embedded at 60 °C for 48 h, after which the resin became polymerized and solid.

Blocks were sectioned using a Reichert–Jung Ultracut E ultramicrotome (Reichert-Jung AG, Vienna, Austria), and ultrathin sections (60–90 nm) were collected on pioloform-coated copper grids (AgarScientific Ltd., Stansted, UK). Each block was sectioned at three different depths, i.e., superficial, intermediate, and deep, with each block derived from one POSs supplementation treatment. Sections were counterstained first in 2% uranyl acetate for 10 min, washed briefly twice in distilled water, then in Reynold’s lead citrate for 5 min, again followed by two brief washes in distilled water. The sections were examined using a transmission electron microscope (Philips EM400T; Eindhoven, Netherlands) operated at 80 kV accelerating voltage. Intracellular granules were then counted in 10 whole cells for each treatment (with the 10 cells distributed over the three depths of the same block). These were counted on the microscope during imaging, allowing magnification as required to differentiate between mitochondria and other granules. Representative images were taken for each treatment. Two types of granules were encountered: granular and homogeneous (more lipofuscin-like), and they were counted separately.

### 2.14. Statistical Analysis

Statistical analyses, including one- or two-way ANOVA followed by pairwise comparisons by Holm–Sidak and Tukey tests, were carried out using SigmaPlot 14.

## 3. Results

### 3.1. Partial Oxidation of 22:6PC

Many natural lipids show an absorption maximum in the UV range close to 200 nm [66]. To avoid an overlap with the solvent absorption, the absorbance of the lipophilic extract from 22:6PC liposomes was monitored at 210 nm and showed a monotonic decrease with the oxidation time of liposomes, corresponding to the loss of docosahexaenoate (Figure 1A,B). The initial docosahexaenoate loss was accompanied by an increase in absorbance at 230–240 and 270 nm. Oxidation of docosahexaenoate can be expected to result in the formation of products with conjugated diene structures, which absorb light at approximately 230 nm [66,67,68], and also trienes and ketone dienes, which exhibit absorption maxima at 260–280 nm [56,66,69,70]. Both types of such absorption maxima can be seen in the absorption spectra of partly oxidized 22:6PC (Figure 1). Conjugated diene and triene levels appear to plateau and then decrease with a prolonged duration of oxidation, presumably due to their degradation and formation of other products [56]. The absorbance maxima at ~270 nm of conjugated trienes and other products may overlap with that of conjugated tetraenes, which show broad characteristic absorbance peaks at ~300 nm [71,72].

Following these observations on the first batch of liposomes, the second batch of liposomes, which were used in further experiments, was allowed to oxidize for a considerably shorter period (18 days vs. 175 days) to achieve a 25–30% loss of docosahexaenoate. Absorbance at 210 nm can indicate the upper limit of unoxidized docosahexaenoate present in samples, but it can overestimate its value if there is a contribution to the absorption at that wavelength by oxidized 22:6PC. In an attempt to overcome this problem, docosahexaenoate was hydrolyzed from phosphatidylcholine, followed by extraction in hexane and HPLC to separate docosahexaenoate from its oxidation products. Based on HPLC data, 22:6PC oxidized for 18 days contained about 70% of intact docosahexaenoate.

Over the oxidation time, both the lipid film and liposomes developed a yellow color, which evolved into a darker yellow and then orange. Therefore, in addition to monitoring changes in the absorption of UV light as an indicator of peroxidation, absorption spectroscopy was used to monitor the absorption of visible light at wavelengths above 400 nm in concentrated samples of partly oxidized 22:6PC. As the anterior eye segment in the adult human blocks most ultraviolet light, observation of products able to absorb light of wavelength longer than 390 nm is required to validate the theory that oxidized 22:6PC may act as a major photosensitizer in the typical adult retina [52,73].

### 3.2. Effects of Partly Oxidized 22:6PC on RPE Cell Viability

#### 3.2.1. Effect of Partly Oxidized 22:6PC on RPE Cell Metabolic Activity

When the MTT assay was carried out immediately after exposure, there was a concentration-dependent loss of metabolic activity both in the dark and upon exposure to light (Figure 2A). Increasing the length of time between the exposure of cells to partly oxidized 22:6PC and MTT assay to 24 h decreased the metabolic activity in both light-exposed and dark-maintained cells, particularly for concentrations greater than 0.2 mM (Figure 2). Twenty-four hours post-exposure to 1 mM partly oxidized 22:6PC, the metabolic activity decreased to 47.2 ± 12.9% and 5.2 ± 4.5% in cells maintained in the dark and in cells exposed to 15 mW/cm^2^ light, respectively; the difference between the means was statistically significant (*p* < 0.001). The partly oxidized 22:6PC at 2 mM decreased the metabolic activities even more to (5.2 ± 3.1)% and (1.8 ± 4.1)% in cells maintained in the dark and in cells exposed to 15 mW/cm^2^ light, respectively; the difference between the means was not statistically significant. Clearly, partly oxidized 22:6PC was highly toxic to cells, and its toxicity increased in a dose-dependent manner and was exacerbated by exposure to visible light.

#### 3.2.2. Effects of Oxidized Docosahexaenoate on RPE Plasma Membrane Integrity

Propidium iodide cannot normally cross the intact cell membrane of live cells, and it is only taken up by cells showing loss of membrane integrity; therefore, it is commonly used as a marker of necrotic cell death under conditions that exclude late stages of apoptosis. Propidium iodide uptake can also indicate an alteration in the membrane structure or function due to direct interaction with oxidized lipids. After entering the cell, propidium iodide binds to DNA, thus labeling nuclei of cells with compromised membranes. The uptake of propidium iodide and nuclear labeling observed in ARPE-19 cells treated with partly oxidized 22:6PC and light for 1 h either immediately or 2 h after the exposure was minimal (Figure 3). While the average percentage of cells exposed to 2 mM partly oxidized 22:6PC and light that were stained by propidium iodide 2 h after exposure was 3.1%, there was no significant difference between this mean and the mean percentage of propidium iodide-labeled nuclei after other treatments (ANOVA: *p* = 0.291), thereby indicating that partly oxidized 22:6PC did not affect the integrity of the cell plasma membrane and cells did not undergo necrotic cell death within this time.

#### 3.2.3. Apoptotic Changes: Nuclear Condensation and DNA Cleavage

To determine whether or not the delayed form of cell death was due to apoptosis, the terminal dUTP nick-end labeling (TUNEL) assay for detection of free 3′-OH termini formed as a result of enzymatic cleavage of DNA into 180 base pair fragments, which is a characteristic biochemical marker of apoptosis [63]. Moreover, the cell monolayer was assessed for morphological features of apoptosis, including condensation of nuclei.

Twelve hours after exposure to partly oxidized 22:6PC and light, changes in cell nuclei were observed, with an increased incidence of condensed nuclei and positive TUNEL (Figure 4). Additionally, the cell density appeared to decrease in a dose-dependent manner following exposure to an increased concentration of partly oxidized 22:6PC. However, there was no statistically significant change in total cell number per imaged area or in the prevalence of TUNEL-stained nuclei (ANOVA: *p* > 0.05 for both). The number of cells with condensed nuclei did, however, increase significantly (ANOVA: *p* = 0.0004), from <1% after exposure to light without partly oxidized 22:6PC to ~25% (*p* < 0.001) and to 14.5% (*p* < 0.05) with partly oxidized 22:6PC originating from 1 and 2 mM 22:6PC, respectively. Additionally, the total number of cells showing no indicators of apoptosis (i.e., normal nuclei in Figure 3B) significantly decreased (ANOVA: *p* = 0.021): 31% and 45% reduction after exposure to 1 and 2 mM partly oxidized 22:6PC, respectively (*p* < 0.05 in both cases) (Figure 4B). None of the cells remaining in the monolayer exhibited nuclear fragmentation. Thus, it may be suggested that condensation of the nucleus can be observed while apoptotic cells are still attached and cells with DNA fragmentation are already lost from the monolayer.

### 3.3. Effects of Antioxidants on the Toxicity of Partly Oxidized 22:6PC

Zeaxanthin and α-tocopherol are components of the retina of dietary origin, which are both potent scavengers of free radicals, and zeaxanthin is an efficient physical quencher of singlet oxygen and excited states of photosensitizers, including excited triplet states of oxidized docosahexaenoate [52,60,74,75]. Effects of zeaxanthin or α-tocopherol incorporated into oxidized 22:6PC liposomes were tested under conditions where partly oxidized 22:6PC caused a reduction in metabolic activity to 79% and 61% in dark- and light-exposed cells, respectively (Figure 5). Zeaxanthin not only showed no protective effect against (photo-)toxicity of partly oxidized 22:6PC but, at the highest concentration tested of 4 μM, significantly decreased the metabolic activities to 66% and 41% in dark and light-exposed cells, respectively (*p* < 0.05). In contrast, α-tocopherol was able to ameliorate the toxicity of partly oxidized 22:6PC in a dose-dependent manner, with 20 and 40 μM α-tocopherol able to significantly increase metabolic activity compared with partly oxidized 22:6PC alone (*p* < 0.01 and <0.001, respectively) to values greater than 90%, such that they were no longer significantly different to control levels (*p* > 0.05).

Previous studies have shown that zeaxanthin is susceptible to degradation, and α-tocopherol can inhibit its degradation, thereby allowing it to act for longer as a singlet oxygen quencher, and as a combination, they can offer synergistic antioxidant protection and protect ARPE-19 cells from phototoxicity [57,60,76,77]. Therefore, further experiments were carried out using a mixture of zeaxanthin and α-tocopherol (Figure 6). To minimize zeaxanthin degradation in liposomes before exposure to cells, it was added to liposomal suspension as a solution in DMSO immediately before the exposure rather than incorporated into liposomes during liposomal preparation. DMSO is an effective free radical scavenger [78], so, not surprisingly, DMSO alone offered substantial protection from partly oxidized 22:6PC toxicity, and a 3-fold higher concentration of partly oxidized 22:6PC was required to induce substantial toxicity (Figure 6).

In the absence of α-tocopherol, zeaxanthin underwent rapid degradation during 1 h incubation with oxidized 22:6PC liposomes in the dark, and it degraded even faster when the incubation took place under 15 mW/cm^2^ light (Figure 6A). The rapid degradation of zeaxanthin could explain the deleterious effects of zeaxanthin in the experiments described above because it has been shown that the degradation products of carotenoids can be damaging to nucleic acids and proteins and be cytotoxic [79,80,81]. In the presence of α-tocopherol, the absorption spectrum of zeaxanthin extracted from liposomes with partly oxidized 22:6PC was similar to that of untreated zeaxanthin even after 1 h exposure to light (Figure 6A). Interestingly, neither α-tocopherol nor zeaxanthin on their own nor their combinations were effective in protection against toxicity or phototoxicity of partly oxidized 22:6PC under these conditions. There were no statistically significant differences between the mean metabolic activities of cells exposed to partly oxidized 22:6PC in the absence and presence of individual antioxidants or their combinations (Figure 6B).

### 3.4. Effect of Supplementation of Cells with POSs Enriched in Partly Oxidized 22:6PC on Metabolic Activity and Formation of Lipofuscin-like Granules

#### 3.4.1. Effect of Supplementation of Cells with POSs Enriched in Partly Oxidized 22:6PC on Metabolic Activity

One of the roles of RPE cells, which is essential for the proper structure and function of photoreceptor cells, is daily phagocytosis of shed tips of POSs followed by their lysosomal degradation. It has been shown that oxidative modifications of phagocytosed POSs can lead to their incomplete lysosomal degradation, resulting in the formation of intracellular fluorescent deposits known as lipofuscin [33,82,83,84,85].

To determine whether partly oxidized 22:6PC can affect the formation of lipofuscin-like material from phagocytosed POSs, the cellular fluorescence and accumulation of lipofuscin-like material can be monitored by flow cytometry and TEM. However, to assess the effect of partly oxidized 22:6PC on lipofuscin formation, it is important to ensure that supplementation with POSs containing oxidized lipids does not affect cell viability. Therefore, the MTT assay was used to evaluate the metabolic activity of cells supplemented with POSs with and without different concentrations of partly oxidized 22:6PC (Figure 7). The highest concentration of partly oxidized 22:6PC that was used to enrich POSs was 20 µM; therefore, it was over 10-fold lower than concentrations, causing partial loss of cell viability as a result of acute exposure (Figure 2). The cells supplemented with such POSs thrice per week for a period of two and three weeks showed significant differences between different supplementation regimens. For cells supplemented for two weeks, there was a significant 20% increase in metabolic activity after feeding with POSs with 10 µM partly oxidized 22:6PC (*p* < 0.001) and a significant 14% decrease after feeding with POSs with 20 µM partly oxidized 22:6PC (*p* = 0.003), compared with control cells.

When comparing the effects of the same supplement at two and three weeks, there was a statistically significant increase in metabolic activities of cells supplemented with POSs and 5 µM partly oxidized 22:6PC (*p* = 0.006) between 2 and 3 weeks, whereas cells supplemented with POSs and 20 µM partly oxidized 22:6PC exhibited a statistically significant decrease in metabolic activity. There were no statistically significant differences between supplementation of cells for 2 and 3 weeks when cells were supplemented only with POSs or POSs in the presence of 10 µM partly oxidized 22:6PC.

After 3 weeks, all supplemented groups varied to a greater extent from the control group, with all paired comparisons showing significant differences and metabolic activities increasing to 115%, 121%, and 128% for supplementation with POSs without added partly oxidized 22:6PC, POSs in the presence of 5 and 10 µM partly oxidized 22:6PC, respectively, and decreasing to 62% for POSs with 20 µM partly oxidized 22:6PC (*p* = 0.007 for comparison of controls with cells fed POSs only, and *p* < 0.001 for comparisons of controls with cells fed POSs with different concentrations of partly oxidized 22:6PC).

#### 3.4.2. Effect of Supplementation of Cells with POSs Enriched in Partly Oxidized 22:6PC on the Formation of Lipofuscin-like Granules

The characteristic feature of all lipofuscins is fluorescence when photoexcited with ultraviolet or blue light [33]. Therefore, cell fluorescence was monitored by flow cytometry as a function of the duration of supplementation with POSs in the absence and presence of partly oxidized 22:6PC (Figure 8A). Supplementing cells with POSs with or without partly oxidized 22:6PC caused significant increases in cellular fluorescence at all the time points investigated, i.e., 1, 2, 3, and 4 weeks in comparison with the control cells (*p* < 0.001 for all; Figure 8A). At all time points, supplementing cells with POSs significantly increased fluorescence in comparison with control approximately 6-fold, with no further statistically significant increase when POSs were enriched in 5 µM partly oxidized 22:6PC. However, when POSs were enriched in 20 µM partly oxidized 22:6PC, a further significant increase in fluorescence occurred: approximately 1.8-, 2.2-, and 3.4-fold after 1, 2, and 3 weeks, respectively, when compared with supplementation with POSs only. The supplementation with POSs enriched in 20 µM partly oxidized 22:6PC for 4 weeks resulted in massive cell detachment, indicating cell death.

Of note, the cells supplemented with POSs or POSs enriched with 5 µM partly oxidized 22:6PC showed a broader distribution in the fluorescence histograms than control cells or cells supplemented with POSs enriched with 20 µM partly oxidized 22:6PC (Figure 8A).

It has been previously shown that oxidation of docosahexaenoate leads to the formation of products with fluorescence properties [59]. Therefore, to account for the possibility that an increased cellular fluorescence was due to the incorporation of oxidized 22:6PC in cells without actually increasing the cellular content of lipofuscin, the formation of lipofuscin-like granules was evaluated by TEM after two weeks of supplementation with POSs in the absence and presence of 20 µM partly oxidized 22:6PC (Figure 8B). The granules counted per ARPE-19 cell cross-section were classified either as membraneous, likely to be phagolysosomes and autophagolysosomes, or as homogeneous lipofuscin-like granules. The differences in the average membraneous granule numbers between different supplementation regimens were not statistically significant. There were significant increases in the homogeneous granule numbers and in the total granule numbers in cells supplemented with POSs enriched with 20 µM partly oxidized 22:6PC in comparison with control cells: the total granule numbers increased 3.5-fold (*p* < 0.001); the homogeneous granule numbers increased 5.2-fold (*p* = 0.01). There was a statistically significant 1.98-fold increase in the homogenous granule numbers in cells supplemented with POSs enriched with 20 µM partly oxidized 22:6PC in comparison with cells supplemented only with POSs (*p* = 0.005).

## 4. Discussion

### 4.1. (Photo)toxicity of Partly Oxidized 22:6PC

The results demonstrate that partly oxidized 22:6PC, with 70–75% of docosahexaenoate remaining in an unoxidized state, is highly toxic to ARPE-19 cells in vitro, and its toxicity is dependent on its concentration and is exacerbated by the concomitant exposure to visible light. The cells exposed to toxic levels of partly oxidized 22:6PC can retain plasma membrane integrity and active metabolism just after the exposure but later undergo a delayed type of cell death with condensation of nuclei and positive TUNEL indicating at least some involvement of apoptosis in cell death, with almost total loss of metabolic activity 24 h post-exposure to 2 mM partly oxidized 22:6PC. Clearly, the cytoprotective effects of docosahexaenoate cannot counteract the deleterious effects of its oxidation products.

These results are consistent with published reports on the cytotoxicity of docosahexaenoate provided to cells under conditions facilitating its oxidation or on the cytotoxicity of specific oxidation products of docosahexaenoate [44,45,46,47,48,49,50,51]. For example, it has been demonstrated that 10–75 µM docosahexaenoic acid can exacerbate photooxidative damage to RPE cells when exposed to light in the culture medium F12, which is rich in iron and a potent photosensitizer, riboflavin [86]. It has been shown in another study that 0.3–0.5 mM docosahexaenoic acid can exacerbate the cytotoxicity of hydrogen peroxide to RPE cells cultured in iron-rich DMEM/F12 medium [87].

Under the experimental conditions used in our experiments, a monolayer of confluent ARPE-19 cells was exposed to partly oxidized 22:6PC liposomes to mimic the situation in vivo where the RPE apical surface is in direct proximity to POSs. In POSs, docosahexaenoate accounts for 32% of fatty acyl chains in phospholipids and free fatty acids [1]. The concentration of docosahexaenoate in POSs can be estimated based on (i) the concentration of rhodopsin in POS of 3 mM, (ii) rhodopsin to phospholipid ratio of 1:75, and (iii) docosahexaenoate accounting for 30 to 35% of fatty acyl chains in POS phospholipids from human donors from 6–39-year-old to 47–89-year-old groups, respectively [1,88,89]. These estimates give values of 135 mM and 158 mM of docosahexaenoate in POS in young and old retinas, respectively. Moreover, in the retina, RPE processes extend into the POS layer, increasing the surface area of possible interactions. Thus, the concentrations of docosahexaenoate the cultured RPE cells were exposed to in our experiments are much lower than under physiological conditions. It may be argued that the greater concentrations of docosahexaenoate available to RPE under physiological conditions may provide abundant substrate for enzymatic synthesis of neuroprotective derivatives such as 10,17*S*-docosatriene, also known as neuroprotectin D1 (NPD1), because of its neuroprotective properties for brain neurons and ARPE-19 cells [17].

However, the protective effects of docosahexaenoate in experiments on cultured cells were demonstrated at concentrations several orders of magnitude smaller than its physiological concentrations in POS or those used in this study [13,14,15,16,17,90,91,92,93,94]. For example, it has been shown that increasing levels of docosahexaenoic acid in cultured rat retinal neurons by supplementation with its precursor, eicosapentaenoic acid, at a concentration of 3 µM protects photoreceptors from apoptosis induced by oxidants paraquat and hydrogen peroxide. When the conversion of eicosapentaenoic acid into docosahexaenoid acid is inhibited by pre-treatment of neuronal cultures with CP-24879 hydrochloride, a 5/6 desaturase inhibitor, the increase in docosahexaenoic acid and its protective effect are completely blocked [95].

Moreover, it has been shown that a very low concentration of docosahexaenoic acid, such as 50 nM, is sufficient to upregulate the synthesis of NPD1 in ARPE-19 cells [17]. This low concentration of exogenous docosahexaenoic acid or 50 nM NPD1 can effectively inactivate proapoptotic signaling induced by hydrogen peroxide and TNFα by upregulation of anti-apoptotic protein expression: Bcl-1 and Bcl-x_L_, by downregulation of pro-apoptotic proteins: Bax and Bad, and by activation of PP2A phosphatase [17,92]. Activated PP2A dephosphorylates Ser62 in Bcl-x_L_ phosphorylated as a result of apoptosis activation, thereby allowing it to act as an anti-apoptotic protein that prevents the aggregation of Bax into pro-apoptotic oligomers that form ion channels in the mitochondrial membrane, facilitating the release of cytochrome c into the cytosol [92]. As a result of the inactivation of proapoptotic signaling by 50 nM docosahexaenoic acid, or NPD1, the activation of the effector caspase-3 can be substantially reduced. Interestingly, 50 nM docosahexaenoic acid appears to be more effective in preventing ARPE-19 cell death induced by a combination of H_2_O_2_ and TNFα supplemented to serum-starved cells than the equimolar concentration of NPD1, suggesting that derivatives of docosahexaenoic acid other than NPD1 are more protective than NPD1 [17].

Therefore, it appears that even though micro- and submicromolar concentrations of docosahexaenoic acid are sufficient to protect from cytotoxicity induced by oxidants such as paraquat or hydrogen peroxide, 0.14–1.5 mM docosahexaenoate does not offer effective protection from the products of its oxidation.

The finding that partly oxidized docosahexaenoate can be cytotoxic is of great physiological importance. The abundant docosahexaenoate in POSs is under constant threat of oxidation due to high oxygen tension and exposure to light, which can photoactivate photosensitizers such as retinaldehyde, and iron ions [33]. The concentration of the latter increases with age and even more in AMD. AMD is associated with increased oxidative damage as evidenced by the accumulation of carboxyethylpyrroles, which are adducts with proteins of a product of docosahexaenoate oxidation [39,40,41]. Increased accumulation of lipid peroxidation products has also been documented in retinas with Stargardt’s disease, an inherited retinal degeneration that causes vision loss often already in children and young adults [33]. The death of RPE cells in Stargardt’s disease and AMD occurs on the timescale of years or even decades. In our study, cell death followed one-hour exposure to partly oxidized docosahexaenoate. Therefore, it can be argued that the oxidation levels of docosahexaenoate used in our study exceed the levels that accumulate under physiological conditions where POSs are equipped with antioxidant and detoxification enzymes.

It has been shown that normal levels of docosahexaenoate in retinal membranes increase the susceptibility of the mouse retina to light-induced degeneration and accumulation of 4-hydroxyhexenal and other end-products of lipid peroxidation, in comparison with the retina partly depleted of docosahexaenoate [96,97,98]. One hour of exposure to 10 klux fluorescent light leads to a decrease in retinal levels of docosahexaenoate in the wild-type mouse and in a mouse model of Stargardt’s disease *abca4(−*/*−)rdh8(−*/*−)* double knockout mouse and retinal degeneration which is more pronounced in the double knockout than in the wild-type mice [99].

It has been shown in several studies on rodents and quails that retinal levels of docosahexaenoate can be increased by supplementation [9,100,101,102,103,104,105,106]. Interestingly, there are also studies demonstrating that supplementation with fish oil containing docosahexaenoate and its precursor eicosapentaenoate of 7-week-old albino Sprague–Dawley rats or 8-month-old wild-type and *abca4−*/*−* knockout mice results in no change of retinal docosahexaenoate, whereas supplementation of 24-month-old wild-type mice results in a 21% decrease in retinal docosahexaenoate [103,107,108].

It has also been shown that supplementation of mice and rats with docosahexaenoate increases the content of not only docosahexaenoate in the tissues, including brain, retina, and blood plasma, but also the content of products of lipid peroxidation, including 4-hydroxyhexenal [106,109,110,111,112]. Several studies on humans have shown that supplementation with docosahexaenoate is associated with increased markers of lipid peroxidation in blood plasma [113]. Oxidation of docosahexaenoate can occur already in supplements before ingestion [114,115,116,117] or in the gastrointestinal tract [118,119]. It remains an open question whether oxidized docosahexaenoate from blood can be taken up into the retina and/or it can deplete antioxidants and cause a net loss of retinal docosahexaenoate.

The oxidation of docosahexaenoate can explain seemingly conflicting results from epidemiological and clinical studies looking at the association of AMD development and/or progression with docosahexaenoate dietary intake and supplementation [19,22,23,24,29,32,120]. While several studies have shown that increased dietary intake of docosahexaenoate or its precursors is associated with a decreased risk of AMD, some clinical trials testing the effects of supplementation with docosahexaenoate have demonstrated no statistically significant effects on AMD development or its progression to an advanced stage. Moreover, a statistically significant 15% increased rate of incident geographic atrophy progression, measured as a square root of geographic atrophy area, has been observed in patients supplemented with docosahexaenoate and eicosapentaenoate in comparison with patients not supplemented with these fatty acids (*p* = 0.037) [23,24,29,32,120].

The oxidation of docosahexaenoate may also explain the results of two large prospective studies, the Nurses’ Health Study (NHS) and the Health Professionals Follow-up Study (HPFS), showing that high dietary intake of docosahexaenoate (including supplements) is protective for the development of moderate AMD but not for advanced AMD [19]. It can be argued that once AMD reaches the moderate stage, the increased oxidative stress in the retina facilitates the oxidation of docosahexaenoate. Therefore, increased docosahexaenoate intake may be more beneficial as prophylaxis than at a stage of the disease where oxidative stress is increased.

The finding that light exacerbates toxicity of partly oxidized 22:6PC is extremely important for a tissue such as the retina. ARPE-19 cells were exposed for 1 h to visible light of 15 mW/cm^2^ irradiance, including 2.84 mW/cm^2^ irradiance in the spectral range of 400–500 nm that can photoactivate oxidized docosahexaenoate so it can produce reactive oxygen species such as singlet oxygen and superoxide [52]. It has been estimated that the average irradiance levels reaching the retina when the eye is exposed to indirect outdoor sunlight or indoor sources of artificial light vary from approximately 0.01 to 0.1 mW/cm^2^ [121,122]. However, when the image of the Sun at the zenith is focused on the retina, it provides an irradiance of 1.6 W/cm^2^ from the spectral range of 400–500 nm in a small retinal area of just 0.16 mm in diameter [33,123]. This means that to provide the same dose of 400–500 nm light as in our experiments, the image of the Sun would need to be focused on the same spot of the retina just for 6.4 s. While most people would not deliberately gaze into the Sun, there are situations where the image of the Sun could be focused on the same spot of the retina multiple times over a period of several hours, therefore creating a potential scenario when radiant exposure could reach potentially phototoxic levels.

### 4.2. Effects of Antioxidants on (Photo)toxicity of Partly Oxidized 22:6PC

To further understand the pathways by which oxidized docosahexaenoate is (photo)cytotoxic and to determine whether the antioxidants normally present in the retina can protect from these (photo)toxic effects, the abundant retinal antioxidants α-tocopherol and zeaxanthin have been tested at physiologically relevant concentrations, which had been previously shown to be protective for ARPE-19 cells exposed to photodynamic damage involving free radicals and singlet oxygen [57,60,74,77].

While α-tocopherol significantly increases, in a dose-dependent manner, the metabolic activity of cells exposed to partly oxidized docosahexaenoate, both in the dark and during simultaneous exposure to light, zeaxanthin does not provide protection and, at 4 μM concentration, it enhances both toxicity and phototoxicity.

Alpha-tocopherol is a lipid-soluble free radical scavenger that is effective as a chain breaker in lipid peroxidation [34]. Hydrogen donation from the phenol group in α-tocopherol to a lipid peroxyl radical results in the formation of an α-tocopherol phenoxy radical. The rate constant for this reaction is almost four orders of magnitude faster than the interaction between the lipid peroxyl radical and lipids, with both constants estimated for biological membranes [124]. Importantly, α-tocopherol phenoxy radical is relatively unreactive with other lipids and oxygen; therefore, it needs to accumulate in a relatively high concentration to play a pro-oxidant role by abstracting hydrogen from unsaturated lipids, resulting in lipid radical formation and propagation of lipid peroxidation [125,126,127]. It is also possible for α-tocopheroxyl radicals to form adducts with lipid peroxyl radicals, thereby terminating the lipid peroxidation chain.

In the presence of DMSO, neither α-tocopherol nor zeaxanthin have any significant effect on the metabolic activities of cells exposed to partly oxidized docosahexaenoate, suggesting that 28.2 mM DMSO competes effectively with 0.04 mM α-tocopherol or 0.004 mM zeaxanthin for docosahexaenoate-derived free radicals and/or α-tocopherol/zeaxanthin are used for scavenging DMSO-derived free radicals [128]. Interestingly, α-tocopherol can completely protect zeaxanthin from photodegradation in the presence of partly oxidized docosahexaenoate, whereas DMSO does not.

Zeaxanthin is a member of the carotenoid family, and like many other carotenoids, it can scavenge free radicals and quench photoexcited states of photosensitizers and singlet oxygen [75,129,130]. The singlet oxygen quenching by zeaxanthin occurs mostly by an energy transfer, and it is an efficient process with bimolecular rate constants approaching the diffusion-controlled limits. As a result of the energy transfer, singlet oxygen returns to its ground state, which is a triplet state, and the zeaxanthin triplet state is formed, which dissipates the transferred energy via thermal deactivation and returns to its ground state. We have shown that zeaxanthin can quench the triplet state formed by photoexcitation of oxidized docosahexaenoate [52].

As with α-tocopherol, it has been suggested that carotenoids may be pro-oxidant in certain situations, mainly as a result of their interaction with free radicals. Carotenoid pro-oxidant actions have been attributed to the formation of adducts with peroxyl radicals or to the electron transfer leading to the formation of reactive carotenoid cation radicals and further carotenoid oxidation leading to the formation of reactive carbonyls, which are cytotoxic [79,129,130,131].

The finding that zeaxanthin can exacerbate cytotoxicity in the presence of partly oxidized docosahexaenoate deserves particular attention because zeaxanthin is present in the proximity of docosahexaenoate in POSs, where they are both at risk of oxidation [1,33,74,132,133].

Commercially available supplements often include lutein/zeaxanthin combined together with docosahexaenoate, despite the results of a large Age-Related Eye Disease Study 2 (AREDS2) that have shown that there is no benefit from supplementation of people with moderate AMD with docosahexaenoate and its precursor eicosapentaenoate [22,23,32,120]. Moreover, it has been shown after a 10-year follow-up that the protective effect on AMD progression of lutein and zeaxanthin (*p* = 0.03) disappears when they are supplemented together with docosahexaenoate and eicosapentaenoate (*p* = 0.12) [120].

### 4.3. Effects of Partly Oxidized 22:6PC on the Formation of Lipofuscin-like Deposits from POSs

Firstly, it is important to note that partly oxidized docosahexaenoate at concentrations that are 10-fold lower than the threshold concentration, inducing detectable toxicity upon acute 1 h exposure, can become cytotoxic when administered to cells thrice weekly over a period of 2 or 3 weeks. This suggests that caution should be taken in interpreting results with these concentrations, as some cells may have died and detached, thus potentially leading to an underestimation of the observed effects if the dead cells are the most loaded with oxidatively modified POSs. The cytotoxic effects of POSs enriched with partly oxidized docosahexaenoate may be due to oxidation products such as hydroxydocosahexaenoic acid, which has been shown to cause lysosomal leakage [46].

Supplementing cells with POSs enriched with 20 µM partly oxidized 22:6PC causes an increase in cell fluorescence. Because oxidized docosahexaenoate exhibits fluorescence when exposed to UV or blue light [59], an increase in cellular fluorescence does not necessarily indicate lipofuscin accumulation; it is possible for oxidized lipids to incorporate directly into the cell lipid membranes rather than entering the cells via phagocytosis. Therefore, it has been essential to complement fluorescence measurements with TEM analysis of cell ultrastructure. Results of TEM and fluorescence obtained after 2 weeks of POS feeding consistently demonstrate that enriching POSs with partly oxidized docosahexaenoate increases both fluorescence and lipofuscin-like granule numbers, thereby indicating the increased accumulation of lipofuscin. This is consistent with other findings showing that lipid oxidation can increase the rates of phagocytosis [97,134], thereby potentially overwhelming the degradative capacity of the cell and facilitating the formation of non-degradable products [82].

Our results on increased accumulation of lipofuscin-like material in cultured RPE cells fed with POSs enriched in oxidized 22:6PC are consistent with other studies on increased lipofuscin accumulation as a result of phagocytosis of oxidized POSs or POSs enriched with products of lipid peroxidation, such as malondialdehyde or 4-hydroxynonenal [81,135,136,137,138].

In summary, these results demonstrate that enrichment of POSs with partly oxidized 22:6PC can impair the degradative capacity of RPE cells and result in the increased accumulation of lipofuscin.

### 4.4. Limitations of the Study

This study has several limitations to be extrapolated directly to the physiological human retina in vivo. Firstly, lipid vesicles used in our study were extensively oxidized. Under physiological conditions, the POSs are equipped with enzymes such as glutathione peroxidase 4 (GPX4) and glutathione transferase [139]; therefore, they can reduce lipid hydroperoxides, preventing their decomposition to radicals, each of which can start a chain of lipid peroxidation, and can remove reactive carbonyls formed as a result of lipid peroxidation. Therefore, it can be argued that this level of lipid oxidation could occur only due to acute insults, such as having the image of the sun focused for tens of seconds on the same spot of the retina, with all-*trans*-retinal responsible for the initial formation of lipid hydroperoxides. In such a situation, a relatively short exposure can lead to RPE and photoreceptor death within hours. As mentioned earlier, in retinal diseases, such as Stargardt’s disease and AMD, the death of RPE cells and photoreceptors occurs on the timescale of many years or even decades.

The lengthy procedures of POSs isolation used in our study were likely to deplete their antioxidants before or shortly after they were artificially enriched in oxidized docosahexanoate and supplemented into cultured cells. Therefore, it can be argued that the experimental procedure of POSs isolation facilitated the formation of material no longer digestible by lysosomal enzymes and caused cell death within weeks, which would be rather unexpected under physiological conditions.

Furthermore, ARPE-19 cells used in our study did not have melanin, which, particularly in the young and healthy retina, can sequester transition metal ions such as iron [140]. Therefore, it could be argued that the ARPE-19 cells were more susceptible to lipid peroxidation induced by the decomposition of lipid hydroperoxides by iron ions than RPE cells in the retina under physiological conditions.

## 5. Conclusions

In conclusion, our results demonstrate that partly oxidized docosahexaenoate can be highly toxic to cultured retinal pigment epithelial cells ARPE-19. Its toxicity increases during concomitant exposure to visible light, which involves apoptosis and is not counteracted by the remaining non-oxidized docosahexaenoate. Vitamin E can provide partial protection from docosahexaenoate (photo)toxicity, whereas zeaxanthin not only does not provide protection but undergoes degradation and can increase the (photo)toxicity. Long-term supplementation of RPE cells with POSs enriched with partly oxidized 22:6PC can decrease their viability and increase the accumulation of lipofuscin.

## Figures and Tables

**Figure 1 antioxidants-13-01428-f001:**
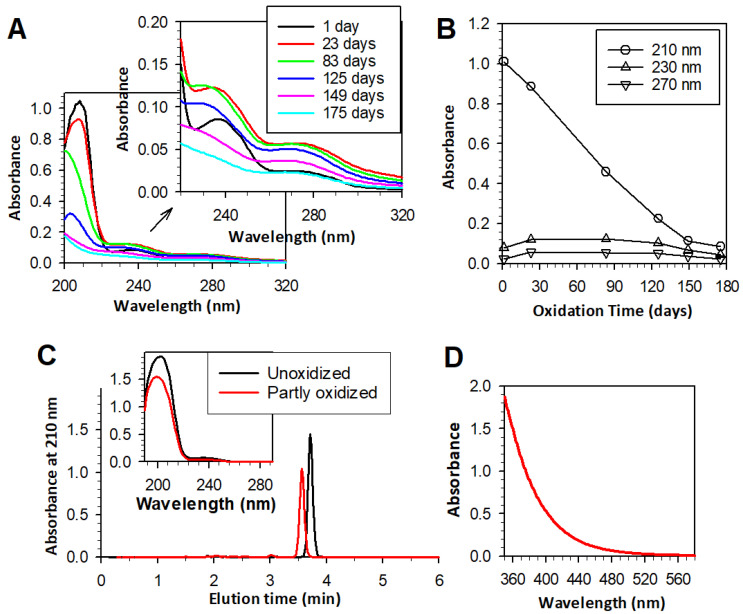
Monitoring oxidation of 22:6PC in liposomes. (**A**): Representative absorption spectra of the chloroform-soluble components extracted from liposomes at indicated times of incubation with air and dissolved in acetonitrile–methanol mixture (20:1 *v*/*v*). (**B**): Changes in absorbance at indicated wavelengths obtained from spectra shown in graph (**A**). (**C**): HPLC chromatograms of docosahexanoate hydrolyzed from PC before and after 18 days of oxidation of 22:6PC as liposomes. Inset: absorption spectra at maxima shown in the chromatogram. (**D**): Absorption spectrum of the 22:6PC oxidation products extracted from the liposomes into the chloroform-enriched phase after 18 days of liposome oxidation, corresponding to 3.75 mM concentration of the original 22:6PC.

**Figure 2 antioxidants-13-01428-f002:**
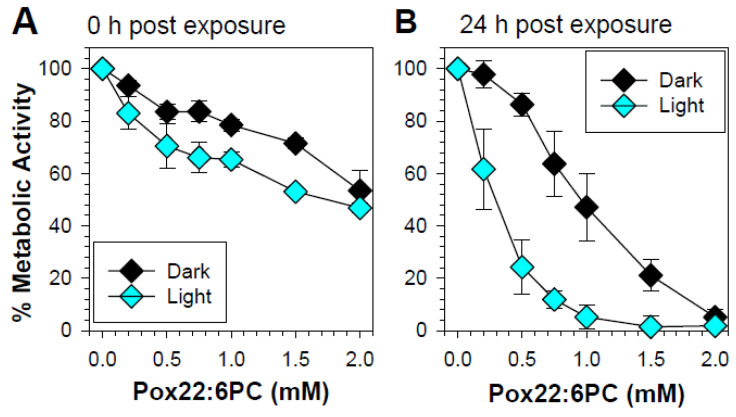
Comparison of metabolic activity of cells measured by MTT immediately after (**A**) and 24 h after (**B**) the exposure to partly oxidized 22:6PC (Pox22:6PC) in dark (Dark) or during concomitant exposure to 15 mW/cm^2^ visible light (Light). Symbols indicate the means; error bars indicate SEM from at least three independent experiments.

**Figure 3 antioxidants-13-01428-f003:**
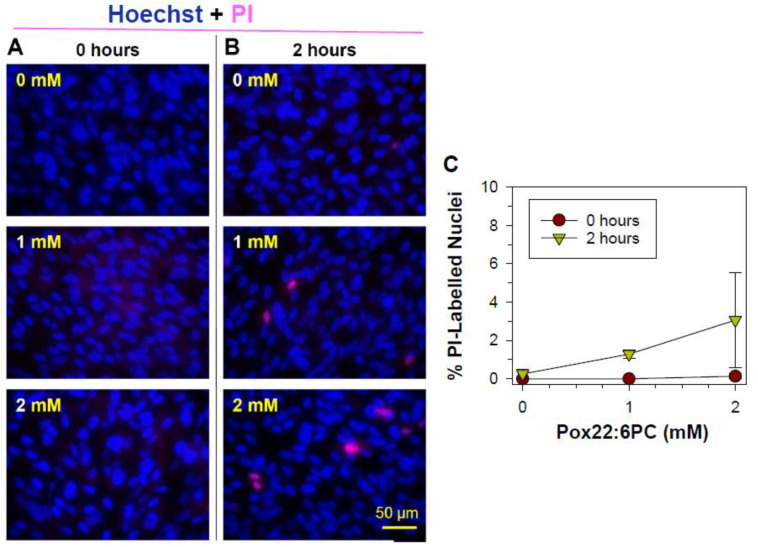
Plasma membrane integrity of ARPE-19 cells immediately after (0 h) and 2 h after 1-h exposure to lipid vesicles containing partly oxidized 22:6PC (Pox22:6PC) at indicated concentrations in the presence of 15 mW/cm^2^ visible light assessed by propidium iodide (PI) assay. Representative images of fluorescence of cells stained with Hoechst (blue) and PI (pink) immediately after the exposure (**A**) and 2 h post-exposure (**B**). All micrographs are taken at the same magnification. (**C**): The ratio of nuclei stained with PI to total nuclei per image expressed as a percentage as a function of Pox22:6PC concentration. Symbols indicate the means, and the error bars indicate SEMs.

**Figure 4 antioxidants-13-01428-f004:**
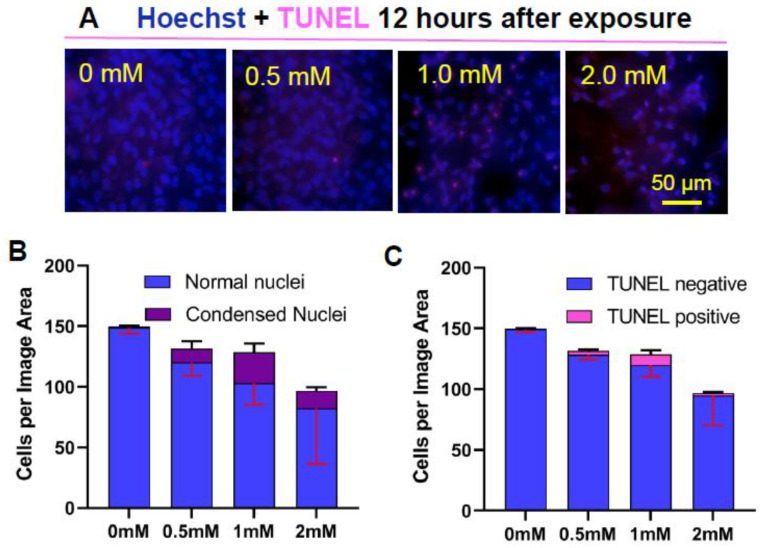
Assays of apoptosis by evaluation of nuclear condensation and TUNEL of cells remaining attached 12 h after the exposure. (**A**): Representative fluorescence images of Hoechst (blue) and TUNEL (pink) of cells fixed 12 h after exposure to indicated concentrations of partly oxidized 22:6PC (Pox22:6PC). All micrographs were taken at the same magnification. (**B**): Quantification of normal and condensed nuclei of remaining attached cells. (**C**): Quantification of TUNEL positive and negative nuclei of remaining attached cells. In (**B**,**C**), the heights of rectangles indicate the means; the error bars indicate SDs.

**Figure 5 antioxidants-13-01428-f005:**
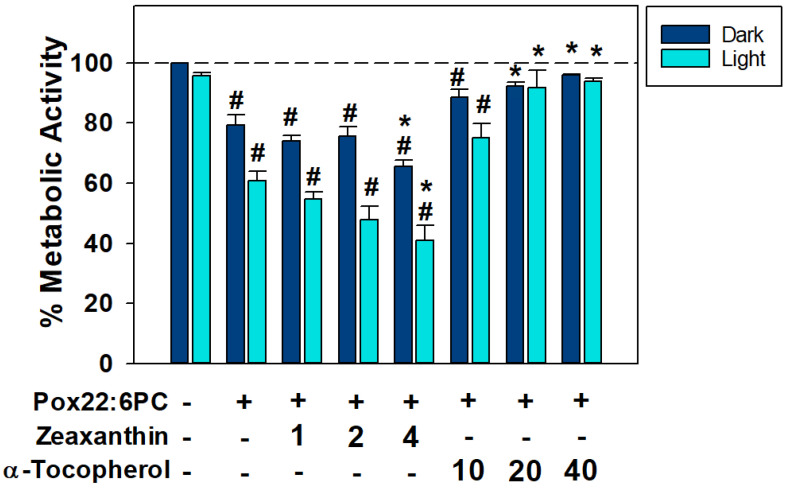
Effect of indicated micromolar concentrations of antioxidants, zeaxanthin or α-tocopherol, on (photo)toxicity of liposomes containing 0.5 mM partly oxidized 22:6PC (Pox22:6PC). (Photo)toxicity was measured by MTT assay of metabolic activity of ARPE-19 cells 24 h after a 1 h exposure to liposomes in the dark or under irradiation with visible light (15 mW/cm^2^). # indicates the mean metabolic activity is significantly different from that of the control cells under corresponding light/dark conditions. The heights of rectangles indicate the means; the error bars indicate SEMs; # indicates the mean metabolic activity significantly different from that of the control cells under the corresponding light/dark conditions; * indicates the mean metabolic activity significantly different from that of the cells exposed to Pox22:6PC without antioxidants under the corresponding light/dark conditions.

**Figure 6 antioxidants-13-01428-f006:**
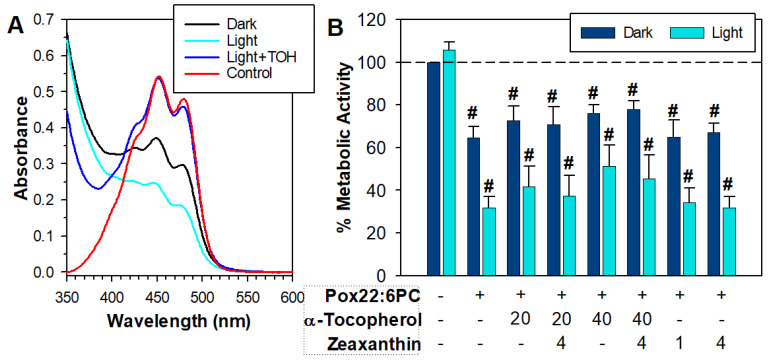
(**A**): Absorption spectra of liposomal extracts obtained after 1-h exposure of zeaxanthin-containing liposomes to light or incubation in the dark in PBS containing 0.2% DMSO. The liposomes contained partly oxidized 1.5 mM 22:6PC (Pox22:6PC) and 4 µM zeaxanthin in the absence or presence of 40 µM α-tocopherol (TOH). The control spectrum of zeaxanthin was obtained by extraction of zeaxanthin delivered into PBS solution directly with DMSO. (**B**): Effect of α-tocopherol and/or zeaxanthin at indicated micromolar concentrations on (photo)toxicity when added to Pox22:6PC liposomes from a solution in DMSO immediately before adding to cells in the culture wells. (Photo)toxicity was measured by MTT assay of metabolic activity of ARPE-19 cells 24 h after 1 h exposure to liposomes in the dark or under irradiation with visible light (15 mW/cm^2^). The heights of rectangles indicate the means; the error bars indicate SEMs; # indicates the mean metabolic activity significantly different from that of the control cells under the corresponding light/dark conditions; there are no statistically significant differences between the mean metabolic activities of the cells exposed to Pox22:6PC in the absence and presence of antioxidants under the corresponding light/dark conditions.

**Figure 7 antioxidants-13-01428-f007:**
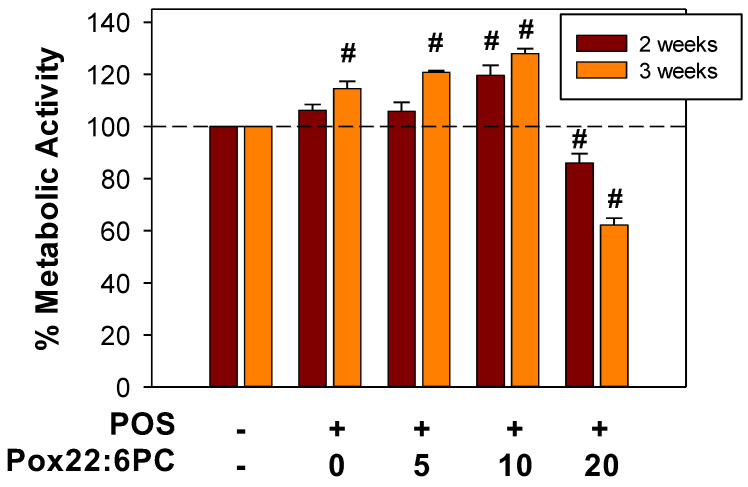
MTT assay of metabolic activity carried out after supplementing cells for 2 or 3 weeks with POSs in the absence or presence of indicated micromolar concentrations of partly oxidized 22:6PC (Pox22:6PC). All treatments were normalized to control cells from the same culture plate. The heights of bars indicate the means; error bars indicate SEMs; # indicates the mean metabolic activity significantly different from that of the corresponding control cells.

**Figure 8 antioxidants-13-01428-f008:**
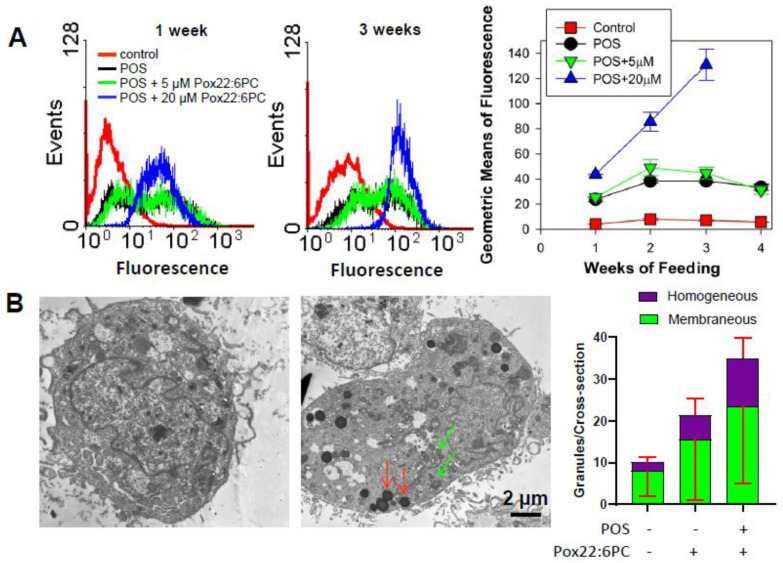
Accumulation of lipofuscin-like material in ARPE-19 cells fed 3 times per week with POSs supplemented with indicated concentrations of partly oxidized 22:6PC (Pox22:6PC) and evaluated by its fluorescence by flow cytometry (**A**) and by transmission electron microscopy (TEM; (**B**)). (**A**): The graphs on the left are representative histograms of fluorescent cells after indicated weeks of feeding; the graph on the right shows the geometric means of fluorescence as a function of time of supplementation with POSs with indicated concentrations of Pox22:6. A total of 10,000 cells was counted for each condition and time point. Fluorescence refers to the intensity of green fluorescence (arbitrary units). Symbols indicate the means; error bars indicate SEMs. Most cells fed POSs supplemented with 20 μM Pox22:6PC (blue lines) died in the 4th week of feeding, so the accumulation of lipofuscin-like fluorescence could not be evaluated at the end of that week. (**B**): on the left, a representative TEM micrograph of a control cell cross-section; in the middle: a representative TEM micrograph of a POS-supplemented cell with green arrows pointing to membraneous deposits and red arrows pointing to the homogenous deposits; on the right: a bar chart depicting the numbers of homogenous and membraneous deposits per cell cross-section after 2 weeks of supplementation with POSs and/or 20 μM Pox22:6PC. The heights of rectangles indicate the means; error bars indicate SDs based on 10 different cells for each treatment.

## Data Availability

All data are included in the manuscript. Digital data are available upon a reasonable request.

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
