# Peer review of "(Photo)toxicity of Partially Oxidized Docosahexaenoate and Its Effect on the Formation of Lipofuscin in Cultured Human Retinal Pigment Epithelial Cells"

_antioxidants, 2024, doi:10.3390/antiox13111428_

Round 1
Reviewer 1 Report
This is an important contribution characterizing in detail the phototoxic and cytotoxic effects of docosahexaenoic acid oxidation on a model of retinal pigment epithelial cells. It has long been known that docosahexaenoate is a major fatty acid present in photoreceptor outer segments and critical for photoreceptor function. At the same time, it is subject to oxidation, which generates cytotoxic and phototoxic products, which have long be considered as potential agents in the development of retinal degenerations, including AMD. The present work provides a solid characterization of the spectrum of damage that can be caused by docosahexaenoate oxidation, along with the potential protection that vitamin E and the carotenoid zeaxanthin may offer, characterization with important clinical relevance.
One potential improvement would be to add a short discussion on considerations of the time scale of the experiments compared to the time scale of the development of human disease. The experiments are of a much shorter duration of course. For example, the authors use the terms lipofuscin and lipofuscin-like for the deposits generated by feeding cells with oxidized docosahexaenoate and photoreceptor outer segment membranes. In the literature, some authors use the term “lipofuscin” to refer to the age-related deposits, and use the term “ceroid” to refer to deposits resulting from acute oxidation. Adding a short discussion on the issue of time scale would be of help to the readers.
Figure 1A is difficult to read -- the colors of the curves shown are too similar. I would recommend changing the colors of the curves.
The numbering of sections in Results is off. There are two sections numbered 3.1, and two sub-sections numbered 3.1.1
Reviewer 2 Report
Not specific work exist related to the photo toxic effect of the oxidized docosahexaenoate. Although imply some points of discussion the work is well done and I think will be intersting for people in the field as well for clinicians.
Small corrections in one or two figures

Round 2
Reviewer 2 Report
The work show important evidence about the effect of oxidized docosahexanoate on lipofuscin production; it is relevant that its adminitration , particulary in elevated dosis may increase oxidative damage to RPE.
Authors followed all my remarks and improve the figure showing lipofuscin deposits